# Sensitization of Patient-Derived Colorectal Cancer Organoids to Photon and Proton Radiation by Targeting DNA Damage Response Mechanisms

**DOI:** 10.3390/cancers14204984

**Published:** 2022-10-11

**Authors:** Kristin Pape, Anna J. Lößner, Doreen William, Tabea Czempiel, Elke Beyreuther, Anna Klimova, Claudia Lehmann, Tim Schmäche, Sebastian R. Merker, Max Naumann, Anne-Marlen Ada, Franziska Baenke, Therese Seidlitz, Rebecca Bütof, Antje Dietrich, Mechthild Krause, Jürgen Weitz, Barbara Klink, Cläre von Neubeck, Daniel E. Stange

**Affiliations:** 1Department of Visceral, Thoracic and Vascular Surgery, Faculty of Medicine Carl Gustav Carus, University Hospital, Technische Universität Dresden, 01307 Dresden, Germany; 2National Center for Tumor Diseases (NCT/UCC), German Cancer Research Center (DKFZ), Faculty of Medicine and University Hospital Carl Gustav Carus, Technische Universität Dresden, Helmholtz-Zentrum Dresden-Rossendorf (HZDR), 01307 Dresden, Germany; 3Core Unit for Molecular Tumor Diagnostics (CMTD), National Center for Tumor Diseases (NCT), Partner Site Dresden, 01307 Dresden, Germany; 4German Cancer Consortium (DKTK), Partner Site Dresden and German Cancer Research Center (DKFZ), 69120 Heidelberg, Germany; 5Hereditary Cancer Syndrome Center Dresden, ERN-GENTURIS, Institute for Clinical Genetics, University Hospital Carl Gustav Carus, Technische Universität Dresden, 01307 Dresden, Germany; 6OncoRay–National Center for Radiation Research in Oncology, Helmholtz-Zentrum Dresden-Rossendorf, Faculty of Medicine and University Hospital Carl Gustav Carus, Technische Universität Dresden, 1307 Dresden, Germany; 7Helmholtz-Zentrum Dresden-Rossendorf, Institute of Radiation Physics, 01307 Dresden, Germany; 8Helmholtz-Zentrum Dresden-Rossendorf, Institute of Radiooncology–OncoRay, 01307 Dresden, Germany; 9Department of Radiotherapy and Radiation Oncology, Faculty of Medicine and University Hospital Carl Gustav Carus, Technische Universität Dresden, 01307 Dresden, Germany; 10National Center of Genetics (NCG), Laboratoire National de Santé, 3555 Dudelange, Luxembourg; 11Department of Particle Therapy, University Hospital Essen, University of Duisburg-Essen, 45147 Essen, Germany

**Keywords:** patient-derived organoid, colorectal cancer, chemoradiotherapy, 3D cell culture, proton radiation, DNA damage response, ATM, translational radio-oncology

## Abstract

**Simple Summary:**

Radiotherapy plays an important role in the treatment of colorectal cancer (CRC). Approximately one-third of patients with rectal cancer show a pathological complete response upon total neoadjuvant treatment. Patient-derived CRC organoids were investigated regarding their radiotherapeutic treatment response—both in terms of conventional photon irradiation, the combination thereof with chemotherapy, as well as proton irradiation. By inhibition of an important sensor molecule for DNA damage, which has been shown to be activated upon irradiation, radioresistant organoids could be resensitized.

**Abstract:**

Pathological complete response (pCR) has been correlated with overall survival in several cancer entities including colorectal cancer. Novel total neoadjuvant treatment (TNT) in rectal cancer has achieved pathological complete response in one-third of the patients. To define better treatment options for nonresponding patients, we used patient-derived organoids (PDOs) as avatars of the patient’s tumor to apply both photon- and proton-based irradiation as well as single and combined chemo(radio)therapeutic treatments. While response to photon and proton therapy was similar, PDOs revealed heterogeneous responses to irradiation and different chemotherapeutic drugs. Radiotherapeutic response of the PDOs was significantly correlated with their ability to repair irradiation-induced DNA damage. The classical combination of 5-FU and irradiation could not sensitize radioresistant tumor cells. Ataxia-telangiectasia mutated (ATM) kinase was activated upon radiation, and by inhibition of this central sensor of DNA damage, radioresistant PDOs were resensitized. The study underlined the capability of PDOs to define nonresponders to irradiation and could delineate therapeutic approaches for radioresistant patients.

## 1. Introduction

For colorectal cancer, 1.9 million new cases and nearly 916,000 deaths were estimated worldwide for 2020, and the incidence is rising [1]. Surgical resection was the mainstay of treatment for patients with localized rectal cancer in the past but is associated with local recurrences. Meanwhile, interdisciplinary approaches including neoadjuvant chemoradiotherapy (CRTx) have significantly decreased local recurrence rates [2,3,4,5,6]. Overall survival was partially improved, dependent on the radiosensitivity of the tumors.

Pathologic regression after neoadjuvant treatment has been associated with better overall survival in several cancer entities and is, therefore, frequently used as an early surrogate marker for survival in clinical trials [7,8,9,10]. For rectal cancer, total neoadjuvant treatment (TNT) with sequential CRTx has been shown to achieve a pathological complete response (pCR) in a substantial fraction of patients (up to 30%) [11,12]. Patients with a pCR can be treated according to a “watch-and-wait” concept, which comprises a close follow-up instead of an operation [13,14,15]. Nevertheless, the remaining 2/3 of the patients do not show a pCR even with TNT. For these patients, all three treatment regimes, namely radio- and chemotherapy followed by surgery, are performed to achieve the best oncological outcome.

While 5-fluorouracil (5-FU)/capecitabin- and oxaliplatin-based chemotherapy as part of TNT regimens often leads to hand–foot syndrome or neuropathy, radiotherapy for CRC is associated with long-term side effects such as incontinence or impotence and dyspareunia [16,17,18]. Surgery for rectal cancer is also associated with significant morbidity and mortality [19]. Thus, the application of only those treatment modalities, which are likely to result in a clinical effect, will lower the overall treatment-associated side effects while resulting in a comparable oncological outcome. 

Besides common photon-based radiotherapy, proton irradiation is gaining significance over the last years. While photons show a gradual loss of their energy within the tissue, proton beams gradually transfer the energy through the tissue culminating in a Bragg peak as the maximum of dose deposition [20,21]. This leads to an improved dose distribution with reduced damage of the surrounding tissue [22]. Therefore, proton-based radiotherapy has the potential to allow a more precise and potentially more effective treatment of rectal cancer. Nevertheless, this needs to be proven in relevant model systems or clinical studies [23,24].

Patient-derived organoids (PDOs) from normal and cancer tissue represent three-dimensional cell cultures with a high degree of similarity to the tissue they are derived from and can be established also from CRC [25]. Living biobanks of CRC PDOs enabled detailed molecular and functional characterization of individual organoids as well as drug screenings to define novel treatment strategies with the hope to facilitate individualization of treatments in the future [26,27,28]. Recent studies from Ganesh et al., Yao et al., and Park et al. focused on PDOs derived from (locally advanced) rectal cancer and could confirm their role as living biomarkers for therapy response [29,30,31]. A clear correlation between the clinical response of the individual patient and ex vivo response of the corresponding PDOs to chemotherapy or photon-based irradiation could be shown. Besides functioning as a biomarker for treatment response, the system has the potential to delineate options to enhance the effectivity of radio- or chemotherapy by implementing strategies to overcome resistance mechanisms of unresponsive cancer cells. Here, we used CRC PDOs to study the effect of photon and proton radiotherapy and showed that the effectiveness can be enhanced by interfering with DNA repair pathways.

## 2. Materials and Methods

### 2.1. Human Tissues

Human colorectal cancer tissues were obtained from patients who underwent biopsies or tumor resections at the Department of Visceral, Thoracic and Vascular Surgery at the University Hospital Carl Gustav Carus of the TU Dresden. Clinical data are summarized in Appendix A. The study was approved by the ethical committee of the TU Dresden (EK451122014), and written informed consent was preoperatively obtained from each patient.

### 2.2. Establishment and Culture of Human CRC Organoids

Approximately 1 cm^3^ of tumor was cut into small pieces and washed with basal Advanced Dulbecco’s modified Eagle medium (DMEM)/F12 (Gibco, Eggenstein-Leopoldshafen, Germany) supplemented with Primocin 1× (Invivogen, Toulouse, France), Glutamax 1× (Thermo Fisher, Waltham, MA, USA), and HEPES 10 mM (Thermo Fisher) until the supernatant was clear. The tissue was digested with 1 mg/mL Dispase II (Roche, Basel, Switzerland) and 0.1 mg/mL Collagenase XI (Sigma-Aldrich, St. Louis, MO, USA) at 37 °C. Regular gentle inversion every 5 min and frequent observation of tumor disaggregation were performed until small tumor patches became visible. Around 150 tumor patches were picked under a stereomicroscope, centrifuged (5 min, 200× *g*), resuspended in 20 µL Matrigel (Corning, Corning, NY, USA), and overlaid with a human colon medium [25] supplemented with 10 µM ROCK inhibitor Y-27632 (Sigma-Aldrich). The established human CRC organoids were cultured as described earlier [25] by passaging once a week with a split ratio of 1:4.

### 2.3. Next-Generation Sequencing

The DNA of the organoids was isolated via phenol–chloroform extraction and subsequent isopropanol precipitation according to standard protocols. Whole-exome sequencing with a median coverage of 150 reads was performed on a Nextseq 500 platform using either IDT xGen Exome Research Panel v1.0 (DD47, DD204, DD254; Illumina, San Diego, CA, USA) or a SureSelect XT HS human all exon V5+UTRs kit (DD72; Agilent, Santa Clara, CA, USA) for library preparation. 

Whole-genome sequencing data were available in FASTq format for DD142 (median coverage of 50 reads). Sequencing data were processed using CLC genomics workbench v12.0 (Qiagen, Venlo, The Netherlands). Reads were trimmed for quality and aligned to the GRCh37 reference genome. Somatic variants were identified by subtraction of variants present in the bam file of matching blood samples. Copy-number variations were called using CNVkit [32] applying circular binary segmentation [33] and GC content bias correction.

Mutational signatures were identified using maftools [34] performing frequency matrix generation and non-negative matrix factorization, as well as following comparison with known mutagenic processes [35,36].

### 2.4. Treatment with (Chemo-)Radiotherapy and Inhibitors

For therapeutic studies, five different organoid lines were plated 24 h prior to their first treatment (Appendix A). Radiotherapy with conventional X-rays was performed in plane position with a dose rate of ~1 Gy/min (Maxishot 200Y.TU/320-D03, Yxlon Int. GmbH, Hamburg, Germany; 200 kV, 20 mA; 0.5 mm Cu filter). 

Proton irradiation took place at the horizontal fixed-beam beamline in the experimental hall of the University Proton Therapy Dresden. A dedicated double-scattering setup for 150 MeV proton beams [37] was applied to form a laterally homogenous proton field of 10 × 10 cm² with a 2.6 cm spread-out Bragg peak. The 2D dose distribution of the unaffected irradiation field was controlled daily using a Lynx scintillation detector (IBA Dosimetry GmbH, Schwarzenbruck, Germany) at the sample position. For irradiation, 96- or 384-well plates with PDOs were positioned upright in the middle of the spread-out Bragg peak using a range shifter of 11 polycarbonate plates (36). Daily dosimetry was performed as described earlier [38] and included the cross-calibration of the monitor ionization at the beam exit (model 34058, PTW) to a Markus chamber (model 23343, PTW) at the sample position. All organoid irradiations were performed at ~3 Gy/min. Proton doses were applied as physical doses without correction for relative biological efficiency.

For combined CRTx studies, the chemotherapeutics 5-FU (Medac, Wedel, Germany), oxaliplatin (Sanofi, Paris, France), and irinotecan (Amneal, Bridgewater Township, NJ, USA) were added 24 h after seeding, and after 8 h, the cells were irradiated (Appendix A). For the ATM inhibition, 24 h after seeding, the organoid medium was supplemented with AZ32 or KU-55933 (Selleckchem, Houston, TX, USA), and radiation was conducted 3 h later. After four days, the medium was changed in all experiments with renewed inhibitors or chemotherapeutics.

### 2.5. Viability Assay

For treatment studies, organoids were initially plated as triplicates in 96-well plates (with 50 µL Matrigel and a 150 µL medium) or 384-well plates (with 15 µL Matrigel and a 50 µL medium). The cell viability of treated PDOs was analyzed after seven days by using a Presto Blue Cell Viability Reagent (Invitrogen). The Presto Blue reagent was mixed with a human colon medium in a ratio of 1:10 according to the manual, and PDOs were overlaid by the mixture. After incubation for 3 h at 37 °C, the fluorescence was measured (Varioscan Lux, Thermo Scientific). Relative viability was normalized to the respective untreated control per PDO. The areas under the curve (AUCs) per drug response curve were determined by GraphPad Prism (version 9.3.1, GraphPad, San Diego, CA, USA) and normalized to the expected AUC if the viability would be constant at 100%.

### 2.6. DNA Damage Repair Assay

Organoids were plated in 48-well plates with 5 wells per sample, each with 20 µL Matrigel and a 250 µL human colon medium. One hour and 24 h after irradiation, PDOs were harvested and digested with TrypLE (Life Technologies, Carlsbad, CA, USA) for 15 min at 37 °C (Appendix A). After washing, cells were fixed with 4% formaldehyde overnight at 4 °C, following permeabilization with 0.2% Triton-X100 (Sigma-Aldrich)/1× PBS for 20 min. After blocking with 1% BSA/1× PBS for 1 h at room temperature, cells were stained with γH2AX (05-636, 1:1000, Merck Millipore, Darmstadt, Germany) and 53BP1 antibody (NB100-904, 1:2000, Novus Biologicals, Wiesbaden, Germany). As secondary antibodies, Alexa Fluor 594 and Alexa Fluor 488 were used, and 4′,6-diamidin-2-phenylindol (DAPI) was used as the nuclear stain (1:2000). Cells were embedded with a Vectashield Mounting medium (Vector laboratories, Newark, CA, USA). γH2AX- and 53BP1-positive foci were manually scored per intact cell nucleus using a Leica DMI 3000 B microscope with 63× objective (Leica, Wetzlar, Germany). The average number of foci per nucleus was calculated from a minimum of 50 nuclei per dose and time point. Normalized residual foci were determined as foci_24h_ minus foci_1h_, and linear regression (through the origin of the coordinate system) was performed. The slopes could finally be correlated with the viability assays.

### 2.7. Western Blot Analysis

Organoids were washed and lysed in a RIPA buffer (Appendix A). Twenty micrograms of denatured protein lysate was separated using the NuPAGE SDS-PAGE Gel System (Invitrogen, Carlsbad, CA, USA) and transferred to a nitrocellulose membrane (Hybond ECL) for 2 h. After blocking, the membranes were probed with the following primary and secondary antibodies (all from Cell Signaling Technology, Cambridge, UK):Phospho-ATM (Ser1981) (D6H9) rabbit monoclonal antibody #5883;ATM (D2E2) rabbit monoclonal antibody #2873;GAPDH (14C10) rabbit monoclonal antibody #2118;anti-mouse IgG HRP linked secondary antibody #7076;anti-rabbit IgG HRP linked secondary antibody #7074.

The chemiluminescence could be visualized by an Immobilon Western Chemiluminescent HRP Substrate (Merck Millipore) and detected with G:Box Chemi XT4 (Syngene, Bengaluru, India). For quantification, the integrated density of the bands was analyzed with ImageJ (version 1.53k, W.S. Rasband, ImageJ, U. S. National Institutes of Health, Bethesda, MD, USA). After subtraction of the background and normalization to GAPDH, the ratios of the bands were determined.

### 2.8. Statistical Analysis

Data are presented as mean ± standard deviation. At least three independent experiments were performed for all endpoints. Statistical analysis was performed with R, IBM SPSS Statistics, and GraphPad Prism. *p*-values ≤ 0.05 were considered as statistically significant (* ≤ 0.05, ** ≤ 0.01, *** ≤ 0.001, and **** ≤ 0.0001). Grouped analyses of drug response curves were performed with a two-way repeated-measures ANOVA. The AUCs, foci, and slopes of normalized residual foci were compared by the unpaired two-tailed Student’s *t*-test.

For the approval of the ATM inhibitors having a (biological) effect on radiation treatment [39], the interaction effects were estimated using a linear regression model with log10(viability) as the outcome and main effects of therapy type (0/3 µM AZ32 or 0/10 µM KU-55933) and irradiation (0/6 Gy) and their interactions as the predictors. The following interpretations were applied:Interaction effects < 5% = additivity;Effects between 5% and 10% = marginal additivity;Higher than 10% = antagonism or synergy.

## 3. Results

### 3.1. Genetic Characterization of Patient-Derived CRC Organoids

In order to understand the differences in response to chemotherapy as well as photon and proton irradiation in the subsequent experiments, the mutational landscape of five PDOs from our CRC biobank was characterized by whole-exome sequencing (Figure 1, Appendix A).

Sequencing revealed the typical CRC mutation pattern as described by The Cancer Genome Atlas (TCGA) study (Figure 1a, Appendix A) [41]. The analyzed PDOs contained 89−1620 variants including inactivating mutations of the tumor suppressors *APC* and *TP53* as well as activating mutations of the oncogenes *KRAS*, *NRAS*, or *PIK3CA*. Interestingly, DD72 revealed an ATM (ataxia telangiectasia mutated) p.Leu2427Arg mutation with uncertain significance, and DD142 carried a heterozygous NBN (nibrin) p.Ser369* mutation. Both encoded proteins are important for sensing and repairing of DNA damage. Typical CRC copy-number variations with recurrent gains on 7p, 12q, and 20p/q occurred (encoding, for example, for *BCL2L1* and *SRC*) and losses within 1p, 4q, 8p, 14q, 15q, 17p, and 18p/q (resulting in deletions of, for example, *ARID1A*, *TP53*, and *SMAD4)* (Appendix A).

Analyzing the mutational signatures of the five PDOs, two signatures could be assigned, which showed the highest similarity to single-base substitution (SBS) 1 and SBS6 (cosine similarity: 0.937 and 0.76) (Figure 1b). Both are characteristic for CRC [35,36], whereas SBS1 is associated with spontaneous deamination of 5-methylcytosines, resulting in C>T transitions, and correlates with age, and SBS6 is linked to mismatch repair deficiency.

### 3.2. Heterogeneous Response of CRC PDOs upon Irradiation

The response of CRC PDOs toward radiotherapy was analyzed. As proton therapy has not yet been examined using CRC organoids, the lines were irradiated with both beam modalities. PDOs were plated and irradiated, and viability was measured 7 d post-treatment (Figure 2a, Appendix A). Proton irradiation was conducted at the experimental beamline of the University Proton Therapy Dresden using specific setups [37,42]. Doses are depicted as physical doses without correction for relative biological efficiency.

A heterogeneous response could be documented between the different PDOs based on the slope of viability curves and the maximum decrease in viability (Figure 2b). No correlation could be seen between the growth rate of the PDOs and their response to irradiation (Appendix A). Independent of the type of radiotherapy, the PDOs could be classified into two groups: three PDO lines were radiosensitive (DD47, DD72, and DD204) as they showed a strong viability reduction in a dose-dependent manner. Two PDO lines, DD142 and DD254, showed a significantly smaller reduction of viability in comparison and were thus deemed radioresistant (ANOVA resistant vs. sensitive PDO: *p*_photon_ ≤ 0.0006, *p*_proton_ < 0.0001). Calculating the normalized area under the curve (AUC), the viability of the radioresistant PDOs was reduced by 9.9–18.3% after photon therapy and 9.6–13.4% post proton treatment, whereas the radiosensitive lines showed a reduction to 52.1–64.6% and 47.3–68.5%, respectively (t-test resistant vs. sensitive organoid lines: *p*_photon_ ≤ 0.0005, *p*_proton_ ≤ 0.0005) (Figure 2c). The measured differences in viability could be matched to the morphology (Figure 2d, Appendix A). After 10 Gy, no viable organoids remained from the radiosensitive PDOs for both photon and proton irradiation, whereas the growth of the radioresistant lines was less affected.

### 3.3. CRC PDOs Repair DNA Damage to Different Degrees

To investigate if different DNA repair capabilities could explain the heterogeneous response, γH2Ax (gamma H2A histone family member X phosphorylated on Ser139)/53BP1 foci assays were performed. Irradiation-induced DNA damage 1 h after photon or proton therapy and the repair thereof (24 h) was analyzed by immunofluorescence staining (Figure 3).

The number of initial foci (1 h) increased in a dose-dependent manner up to 43−64 and 31−52 foci after treatment with 6 Gy photon or proton irradiation, respectively (Figure 3b). Of note, the radioresistant line DD142 showed a significantly lower number of initial foci both after photon and proton irradiation compared with the radiosensitive PDOs (*t*-test for 2 Gy and 6 Gy: *p* ≤ 0.0001). This could also be observed in the second radioresistant line DD254 but, interestingly, only after proton treatment (*p*_proton_ < 0.0001).

Within the radiosensitive group, more residual foci remained after 24 h compared with the amount in the radioresistant PDOs: the radiosensitive PDOs DD47, DD72, and DD204 still had around 12–24 or 18–22 residual foci after treatment with 6 Gy photons or protons, respectively, which resulted in normalized repair rates of 66–84% and 52–66%, respectively. The repair rates of the radioresistant PDOs DD142 and DD254 were higher (87–91% and 68–75% for photons and protons, respectively), resulting in 6–11 and 9–12 residual foci. Next, the residual foci were normalized to untreated PDOs (0 Gy), and linear regression was performed (Figure 3c). The resulting slope of the normalized residual foci, constituting a commonly used radiobiological marker, ranged from 1.6 to 2.8 and from 2.8 to 3.3 post photon and proton treatment in radiosensitive PDOs, whereas in radioresistant PDOs, the slope ranged from 0.7 to 1.2 and from 1.4 to 1.9, respectively (*t*-test slopes of radiosensitive vs. -resistant PDOs: *p*_photon_ ≤ 0.0396 (except DD204), *p*_proton_ ≤ 0.0431) (Figure 3c). These slopes of the normalized residual foci were significantly linearly correlated with the viability assays (log_10_AUC; photons: *R*^2^ = 0.9827, *p* < 0.001; protons: *R*^2^ = 0.8092, *p* = 0.0376) (Figure 3d).

### 3.4. CRC PDOs Cannot Be Sensitized to Irradiation by 5-FU

Chemotherapy has been discussed to function as a radiosensitizer. Therefore, it was hypothesized that CRT might render radioresistant PDO lines more sensitive to irradiation. The treatment response of CRC PDOs was thus analyzed toward the chemotherapeutics 5-FU, irinotecan, and oxaliplatin alone as well as the combination of photon irradiation with 5-FU (Figure 4).

The differential dose response curves of the CRC PDOs for 5-FU, irinotecan, and oxaliplatin could be documented for all three drugs (Figure 4a,b). Of note, the two radioresistant PDOs DD142 and DD254 also showed a relative resistance to all three chemotherapeutics, whereas the two radiosensitive PDOs DD47 and DD72 responded better. Of note, the radiosensitive line DD204 showed a drug-specific pattern. The line responded to 5-FU treatment while being relatively resistant to irinotecan and oxaliplatin.

For the radiosensitive lines DD47, DD72, and DD204, the combined treatment of 5-FU and photon irradiation revealed a significant additional decrease in the viability of 23.6%, 21.3%, and 13.6% on average, respectively (ANOVA: irradiated vs. non-irradiated *p* ≤ 0.0088) (Figure 4c). The resulting normalized AUCs confirmed this viability reduction (*t*-test: DD47 and DD72 *p* ≤ 0.0328, DD204 n.s.) (Figure 4d). Of note, radiotherapy alone was already effective in these lines. Concerning the radioresistant lines, 5-FU treatment did not sensitize the PDOs to irradiation: dose response curves revealed only a slight viability reduction (DD142: −7.8%, *p* = 0.0140; DD254: −4.9%, *p* = 0.1150), resulting in no significant changes within the resulting normalized AUCs (*t*-test: *p* ≥ 0.0749).

### 3.5. ATM Inhibitors Can Sensitize Radioresistant CRC PDOs

The γH2Ax/53BP1 foci assays revealed that radioresistant PDO lines repaired DNA damage significantly better (Figure 3). To identify potential underlying mechanisms, the phosphorylation status of DNA repair-associated proteins was analyzed in photon- and proton-irradiated CRC PDO lysates (Figure 5). The ATM kinase is important for sensing DNA damage and responsible for the initiation of DNA repair processes. Western blot analyses revealed that ATM was indeed activated by phosphorylation in irradiated CRC PDOs.

To delineate a potential role of ATM as an involved sensor protein upon radiation-induced DNA damage, PDOs were treated with two potent and specific ATM inhibitors, KU-55933 [43] and AZ32 [44], in conjunction with 6 Gy photon or proton irradiation (Figure 6).

In general, ATM inhibitors alone showed no or minor effects on the viability of PDOs even at the highest doses (Figure 6a, continuous lines). Concomitant treatment of the radiosensitive lines with ATM inhibitors and irradiation led to a slight reduction of the viability for AZ32 (0.7–16.94% at the highest dose, n.s.) and KU-55933 (4.3–28.1% at the highest dose, n.s.). For both radioresistant PDOs, combined treatment with AZ32 or KU-55933 enhanced the effect of irradiation, resulting in a viability reduction up to additional 24.2–45.4% and 28.8–59.7% at the highest dose, respectively (*t*-test *p*_DD142_ ≤ 0.0395, *p*_DD254_ n.s.). The result of the statistical analysis of interaction effects revealed synergy for nearly all combinations of PDOs and the two ATM inhibitors, statistically significant for DD204 and DD142 (Appendix A). In addition, while the AUCs of the drug response curves already showed a strong irradiation-caused viability reduction per se in radiosensitive PDOs, the AUCs of radioresistant organoids were now also significantly decreased by concomitant treatment with ATM inhibitors and radiation (*t*-test *p* ≤ 0.04) (Figure 6b).

## 4. Discussion

The development of the organoid technology has revolutionized in vitro cell cultures. Three-dimensional organoids are grown in an extracellular matrix supplemented with growth factors mimicking their natural stem cell niche, allowing them to retain the heterogeneity and genetic identity of the (cancer) tissue from which they are derived. CRC-derived organoids were the first to be described and for which so-called living organoid biobanks could be established [25,26,27]. Several studies have confirmed that CRC PDOs recapitulate the histopathological and genetic features of their parental tumors [26,27,30,31,45]. In line with this, genetic analyses of the PDO lines used for this study revealed typical mutational patterns and copy-number variations of CRC, such as alterations within the WNT, RAS, TP53, and TGFβ signaling pathways [41,46]. In addition, the mutational signatures SBS1 and SBS6, characteristic for CRCs, could be assigned to the here-analyzed PDO lines [35,36].

Besides being a valuable tool to generate large biobanks representing the whole spectrum of molecular diversity of a cancer entity, individual PDOs function as avatars of the patient´s tumor that might be used to personalize therapy [47]. Three recent studies by Ganesh et al., Yao et al., and Park et al. focused on this aspect and used CRC-derived organoids for therapeutic response prediction [29,30,31]. They could demonstrate a clear correlation of patient response with the ex vivo organoid response for both chemo- and radiotherapy. Within our cohort, we could also demonstrate a heterogeneous response of CRC PDOs to chemotherapy and irradiation. The lines could be divided into two groups of radioresistant (DD142 and DD254) and radiosensitive PDOs (DD47, DD72, and DD204) based on a statistically significant difference in viability. Of note, the radiosensitivity was not associated with previous irradiation of the tumor before PDO generation. This underlines the unpredictability of the therapeutic response and, therefore, the importance of sensitivity screenings for each individual patient.

New radiation treatment technologies are constantly developed also for gastrointestinal tumors [48]. The excellent dose distribution of proton beams reduces the damage of surrounding tissue and enables the use of higher doses for the target region. Pre- and postoperative proton therapy for locally advanced rectal cancer seems to reduce the exposure toward the small bowel, bladder, femoral heads, and pelvic bone marrow [49,50,51,52,53]. To the best of our knowledge, no previous study has used CRC organoids to investigate proton irradiation, yet. For both photon and proton therapy, we could demonstrate a similar heterogeneous response pattern of the investigated PDO lines, dividing them into the radioresistant and -sensitive groups in both settings. It should be mentioned, however, that the chosen endpoints were adapted from drug screens and may not reflect clonogenic cell survival. As this is a major endpoint in radiobiology, further readouts with longer follow-up will add information in future experiments [23]. However, our data show that PDOs are a potent and clinically relevant tool to study treatment response and, thus, will also be very useful to tackle major open questions in the field of translational (proton) radio-oncology [54].

Irradiation directly or indirectly damages the DNA by free radicals, thereby activating diverse signaling pathways, which results in the death of tumor cells via mitotic catastrophe [55]. In order to determine the DNA damage, we performed γH2Ax/53BP1 foci assays. These assays and their translation into the clinic are comprehensively studied [56,57,58]. No differences in baseline foci could be seen. Significant differences could be detected between PDO lines concerning their initial radiation-induced DNA damage and the capability to repair it. The normalized residual foci revealed a significant linear correlation with the viability assays. Clearly, PDOs with better repair rates of the initial DNA damage and less residual foci showed radioresistance resulting in better survival. Therefore, these PDO-based assays might be a powerful tool as a rapid and valid test for radiosensitivity.

Next to documenting the response to radiotherapy, treatments with the chemotherapeutics 5-FU, irinotecan, and oxaliplatin, frequently applied in CRC treatment, were performed. Overall, similar trends regarding sensitivity or resistance were found between radiotherapy and chemotherapy. We hypothesize that this is due to similar effects of radiotherapy and chemotherapy on target pathways such as DNA synthesis and repair as well as cell-cycle control [59], at least during the follow-up period of seven days after irradiation.

The therapeutic effect of 5-FU combined with radiation is still not fully understood, and additive and synergistic effects have been described in vitro [60]. 5-FU and its analogs execute their cytotoxicity by inhibition of the thymidylate synthase. The lack in thymidine 5′ monophosphate and thymidine 5′-triphosphate inhibits DNA synthesis and repair [61]. Consequently, radiation-induced DNA double-strand breaks cannot be sufficiently repaired [62,63]. Accordingly, radiosensitive organoids showed an additional decrease in viability compared with irradiation alone. However, this observation was not seen for the radioresistant PDOs.

Therefore, alternative combinatorial approaches should be found to sensitize radioresistant PDOs. After the establishment of a correlation between the capability to repair radiation-induced DNA damage and radioresistance in CRC PDOs, potential pathways associated with sensing of DNA damage and the repair thereof were examined. For photon as well as proton therapy, an activating phosphorylation of ATM as a sensor protein could be shown for all lines. Of note, DD72 carried a variant for the ATM gene with uncertain significance, which resulted in p.L2427R within the structural relevant FAT domain [64]. Possibly, this had an influence on the DNA damage response making the cells more radiosensitive. Based on these results and the known role of ATM, we functionally tested the role of ATM as a sensor of radiation-induced DNA damage in CRC PDOs. Overall, a synergistic potentiation of irradiation by concomitant ATM inhibition could be shown. Especially the radioresistant PDOs could be resensitized to irradiation, envisioning a potential use of ATM inhibitors in the treatment of radioresistant CRCs. In ongoing clinical trials (phase I/IIa), ATM inhibitors are currently investigated as single agents or in combination with chemotherapeutic agents [65,66,67]. Before initiating radiotherapeutic clinical trials for these inhibitors in radioresistant CRCs, the testing of ATM inhibitors in relevant CRC mouse models should be performed to confirm their potential effectiveness as sensitizers of radioresistant tumor cells in vivo, similar to studies in mouse models for melanoma and glioblastoma [44,68].

## 5. Conclusions

Recent studies have demonstrated the potential of PDOs as a tool to predict therapy response in CRC [29,30,31]. We could show here that the system allows a stratification of patients into responders and nonresponders to photon as well as proton irradiation. Furthermore, γH2Ax/53BP1 foci assays were established to reliably assess radiosensitivity directly after the establishment of the PDOs. In addition, ATM inhibitors were shown to potentially function as promising radiosensitizers for radioresistant CRC patients.

## Figures and Tables

**Figure 1 cancers-14-04984-f001:**
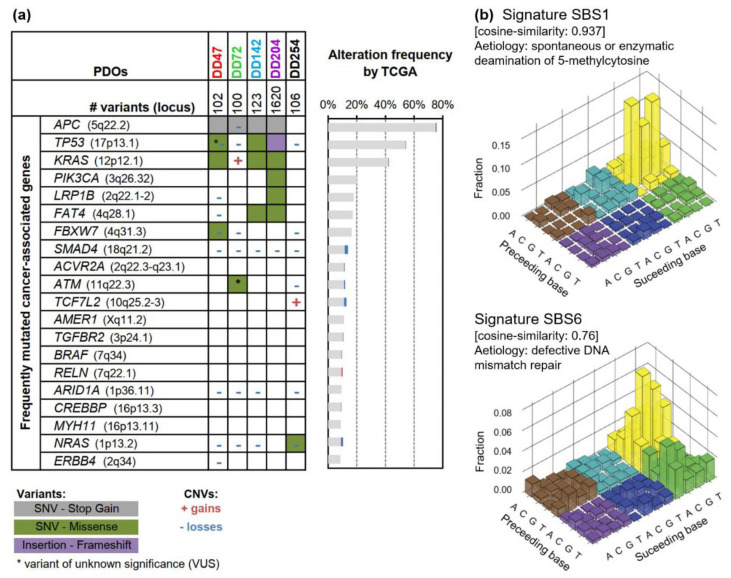
Genetic characterization of CRC patient-derived organoids (PDOs). (**a**) Oncoplot of gene mutations and copy-number variations (CNVs) of PDOs. The 20 most frequently mutated cancer-associated genes (according to the OncoKB database [40]) are listed with their mutation frequency according to The Cancer Genome Atlas (TCGA) CRC dataset (mutation frequency *n* = 224; CNV frequency *n* = 257) [41]. (**b**) Mutational signatures were detected for CRC PDOs, which could be assigned to SBS1 (deamination of 5-methylcytosine, cosine similarity 0.937) and SBS6 (defective DNA mismatch repair, cosine similarity 0.76). SNV: single nucleotide variant; SBS: single-base substitution.

**Figure 2 cancers-14-04984-f002:**
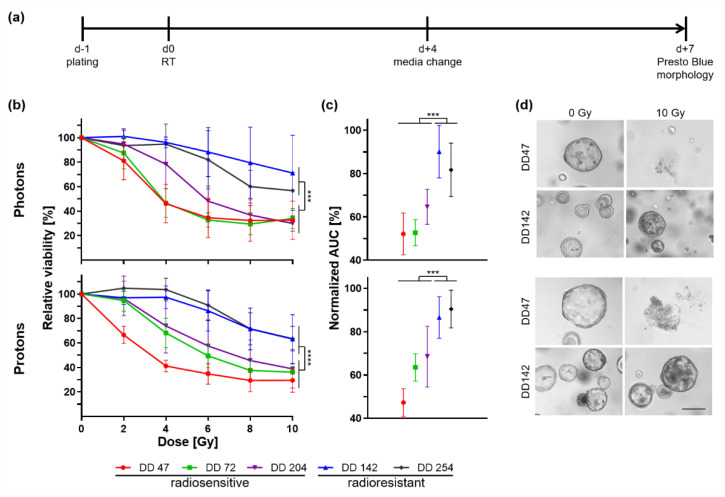
Photon and proton irradiation of CRC PDOs. (**a**) Timeline of experiments. (**b**) Relative viability of irradiated PDOs 7 d post-treatment was measured by a metabolic assay (*p*_photon_ ≤ 0.0006, *p*_proton_ < 0.0001), and (**c**) normalized area under the curve (AUC) was calculated (*p* ≤ 0.0005). (**d**) Representative pictures are shown of organoid morphology upon control (0 Gy) and 10 Gy irradiation of a radiosensitive (DD47) and a radioresistant (DD142) line (scale bar: 200 µM; see also Appendix A). ***: *p* ≤ 0.001, ****: *p* ≤ 0.0001.

**Figure 3 cancers-14-04984-f003:**
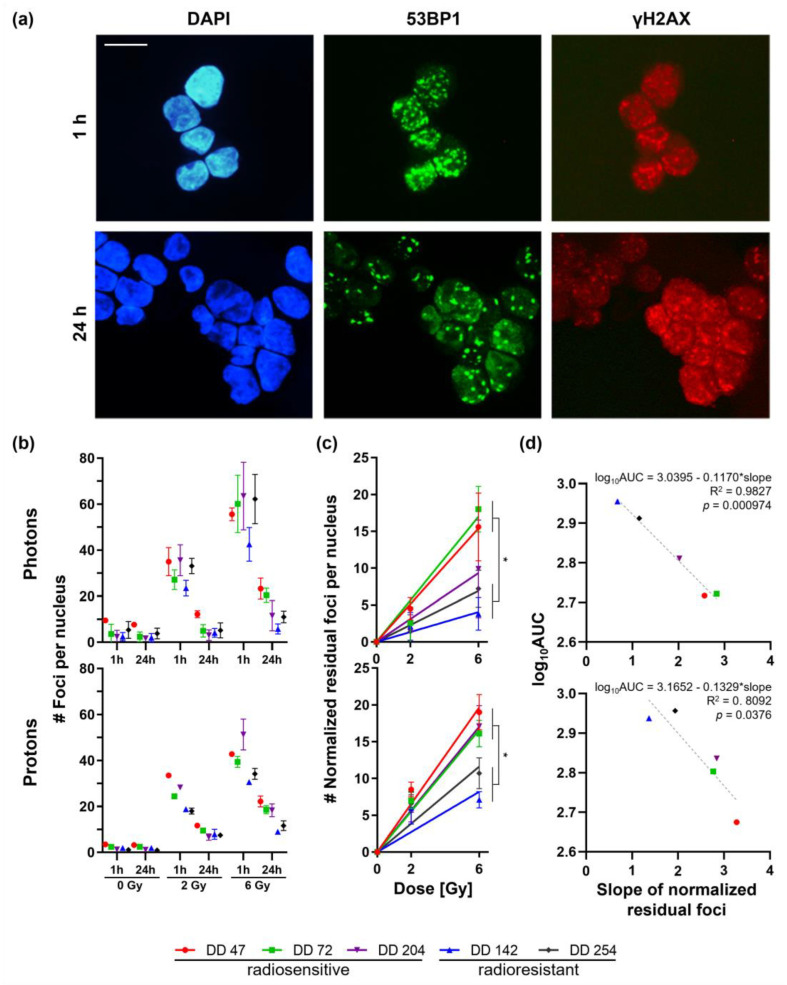
γH2Ax/53BP1 foci assays of irradiated CRC PDOs. (**a**) Representative immunofluorescence stainings of DD142 for γH2Ax/ 53BP1 1 h and 24 h after 6 Gy proton irradiation (scale bar: 10 µM). (**b**) Foci per nucleus 1 h and 24 h post-treatment were counted for control (0 Gy), 2 Gy, and 6 Gy. (**c**) Slopes for normalized residual foci were determined (*t*-test slopes of radiosensitive vs. -resistant PDOs: *p*_photon_ ≤ 0.0396 (except DD204), *p*_proton_ ≤ 0.0431). (**d**) Linear regression of viability (log_10_AUC, see Figure 2c) toward the slope of normalized residual foci revealed significant correlation (photons: R^2^ = 0.9827, *p* < 0.001; protons: R^2^ = 0.8092, *p* = 0.0376). *: *p* ≤ 0.05.

**Figure 4 cancers-14-04984-f004:**
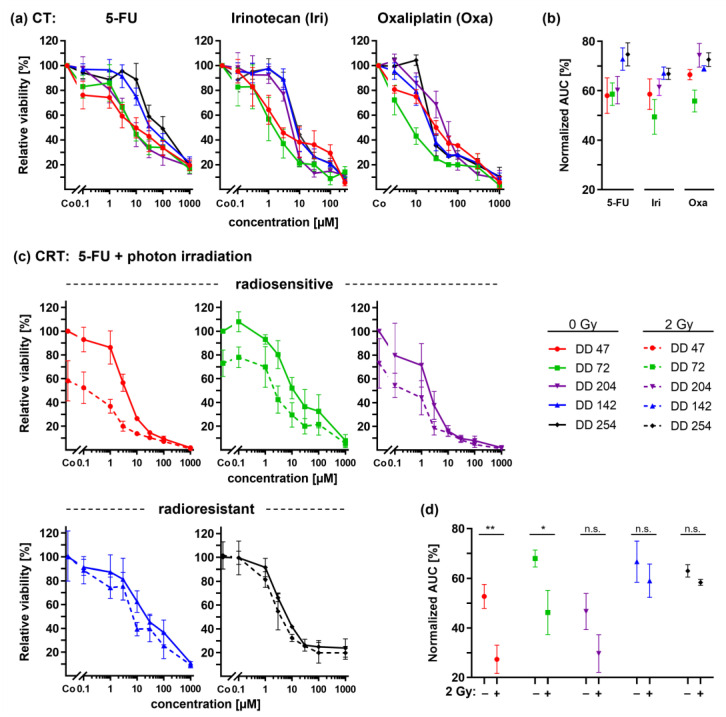
Chemotherapy (CT) and combined chemoradiotherapy (CRTx) of CRC PDOs. (**a**) Dose response curves upon treatment with 5-FU, irinotecan, and oxaliplatin alone and (**b**) resulting normalized AUCs. (**c**) Combined CRTx using 2 Gy photon irradiation with 5-FU and (**d**) resulting normalized AUCs (*p*_DD47_ = 0.0088, *p*_DD72_ = 0.0328). *: *p* ≤ 0.05, **: *p* ≤ 0.01, n.s.: not significant.

**Figure 5 cancers-14-04984-f005:**
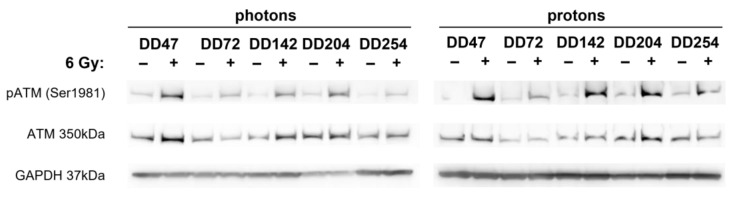
Radiation-induced activation of ATM kinase in CRC PDOs. Western blot analysis of CRC PDOs 4 h post photon and proton irradiation for (phosphorylated) ATM. GAPDH served as loading control.

**Figure 6 cancers-14-04984-f006:**
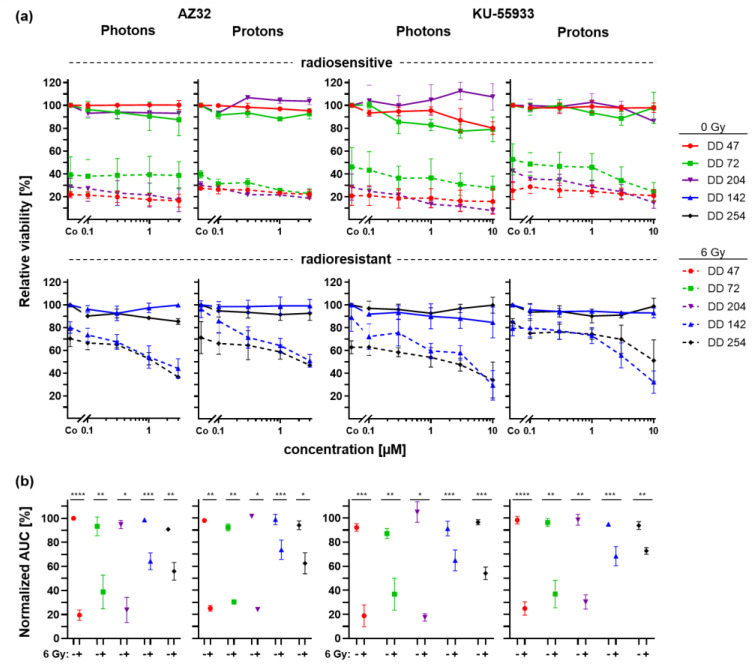
Radiation combined with ATM inhibitor treatment of CRC PDOs. (**a**) Dose response curves of radiosensitive (upper panel) and radioresistant (lower panel) CRC PDOs upon control (continuous line) and 6Gy (dashed line) photon and proton treatment with increasing doses of ATM inhibitors AZ32 and KU-55933. (**b**) Normalized AUCs were calculated. *: *p* ≤ 0.05, **: *p* ≤ 0.01, ***: *p* ≤ 0.001, ****: *p* ≤ 0.0001.

## Data Availability

The data presented in this study are available on request from the corresponding author.

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
