# Peer review of "Sensitization of Patient-Derived Colorectal Cancer Organoids to Photon and Proton Radiation by Targeting DNA Damage Response Mechanisms"

_cancers, 2022, doi:10.3390/cancers14204984_

Round 1

Reviewer 1 Report

This paper examines the use of patient-derived colorectal organoids as surrogates to probe the radio- and chemo-sensitivity of 5 subjects.  New to the field is the comparison of gamma-irradiation and proton beam -irradiation and both with in combination with 5-FU.  Additionally, an attempt is made to overcome apparent radiation resistance of 2 tumor-organoids with DNA damage checkpoint inhibitors.  The study seems to show that the response to proton-beam was not markedly different from gamma-irradiation.  However, some small difference was observed for the damage markers γH2Ax/ 53BP1 between gamma- and proton -irradiation, that were different from the indicated radio-resistance and sensitivity of the organoids.  More interesting was the partial success in sensitizing the resistant lines to radiation with DNA check point inhibition (ATM) given the failure to sensitize the lines with 5-FU.

My complaints center on the small size of the study set and the inability to follow up as in 3 of the referenced studies (29-31), to show any predictive ability of the organoid analysis.  The 3 references were meant to suggests that organoids have good predictive ability, but that is just an argument here.

In any event, the paper has just enough merit to warrant publication.   I trust that this is a pilot study for the group as proof of principle and that a larger study will follow with more rigor, primarily over follow up with patient responses.

For some reason, the uploaded document, triplicates the figures and there are some Reference file not found messages in the text.  Please, sort out the issues.

Author Response

We would like to thank you for your valuable time and useful contribution to our manuscript.

“My complaints center on the small size of the study set and the inability to follow up as in 3 of the referenced studies (29-31), to show any predictive ability of the organoid analysis. The 3 references were meant to suggests that organoids have good predictive ability, but that is just an argument here. In any event, the paper has just enough merit to warrant publication. I trust that this is a pilot study for the group as proof of principle and that a larger study will follow with more rigor, primarily over follow up with patient responses.”

We totally agree with the reviewer. This was indeed an exploratory pilot study, to explore the mechanisms of radiation resistance using CRC organoids. The generated data allows us to design follow-up studies with a larger cohort and related statistical power to demonstrate a possible correlation between organoid and patient response.

“For some reason, the uploaded document, triplicates the figures and there are some Reference file not found messages in the text. Please, sort out the issues.”

We would like to thank the reviewer for pointing out the formatting issue. Some problems with cross references occurred in the manuscript resulting in the duplication of figures or missing links. This has now been solved.

Reviewer 2 Report

The article was well-organized and well-written demonstrating the investigation of the radiotherapeutic treatment response of patient-derived CRC organoids, which found that the radioresistant organoids could get radiosensitization by inhibiting some critical sensors. 

Recommend to publish after minor revisions and give some clarifications on questions as follows: 

1. The article mentioned five PODs were used as starting experimental materials. It's better to specify the five PODs sourced from one individual donor or multiple? and the gender of each patient donor? 

2. Add the animal study data to confirm the effectiveness of radiosensitization in vivo. 

Author Response

We would like to thank you for your valuable time and useful contribution to our manuscript.

“1. The article mentioned five PODs were used as starting experimental materials. It's better to specify the five PODs sourced from one individual donor or multiple? and the gender of each patient donor?”

We thank reviewer 2 for her/his comment. There were some problems with cross references in the manuscript, which resulted in the link to table S1 to be missing. Table S1 lists the patient characteristics of each donor. We have solved the formatting problem and the patient data is now accessible.

“2. Add the animal study data to confirm the effectiveness of radiosensitization in vivo.”

We have not performed animal studies to confirm the effectiveness of radiosensitization in vivo as this is not within the scope for this current study. Nevertheless, we agree with reviewer 2, that confirmative studies are needed. In our view, this should rather be prospective co-clinical trials that compare the organoid response to the in vivo response of the corresponding patient, rather than comparisons to another biological model system (i.e. xenotransplanted mice).

Round 2

Reviewer 1 Report

The authors have addressed by concerns.

Reviewer 2 Report

No further comments